# Bioaccessibility of Antioxidants and Fatty Acids from *Fucus Spiralis*

**DOI:** 10.3390/foods9040440

**Published:** 2020-04-06

**Authors:** João Francisco, André Horta, Rui Pedrosa, Cláudia Afonso, Carlos Cardoso, Narcisa M. Bandarra, Maria M. Gil

**Affiliations:** 1MARE—Marine and Environmental Sciences Centre, ESTM, Polytechnic of Leiria, Portugal; CETEMARES—Centro de I and D, Formação e Divulgação do Conhecimento, Avenida do Porto de Pesca, 2520-630 Peniche, Portugal; andre.horta@ipleiria.pt (A.H.); rui.pedrosa@ipleiria.pt (R.P.); 2Division of Aquaculture, Upgrading and Bioprospection, Portuguese Institute for the Sea and Atmosphere, IPMA, Avenida Alfredo Magalhães Ramalho, 6, 1495-165 Algés, Portugal; cafonso@ipma.pt (C.A.); carlos.cardoso@ipma.pt (C.C.); narcisa@ipma.pt (N.M.B.); 3Interdisciplinary Centre of Marine and Environmental Research (CIIMAR/CIMAR), University of Porto, Rua dos Bragas 289, 4050-123 Porto, Portugal

**Keywords:** seaweed, *Fucus spiralis*, bioaccessibility, antioxidants, fatty acids

## Abstract

*Fucus spiralis* is an edible brown seaweed (SW) found in the Portuguese Coast. It has been reported to have high antioxidant activity, which may elicit a potential use for the food industry. However, little information is available on how the SW behaves during the digestive process and how the freeze-drying process might affect the bioaccessibility of the different compounds. Therefore, antioxidant activity, total polyphenols, lipid, and fatty acid contents were measured before and after in vitro simulation of the human digestive process, both in fresh and freeze-dry SW. *F. spiralis* had a lipid content of 3.49 ± 0.3% of dry weight (DW), which is a usual amount described for this SW genus. The total lipid bioaccessibility was 12.1 ± 0.1%. The major omega-3 fatty acid detected was eicosapentaenoic acid, 7.5 ± 0.1%, with a bioaccessibility percentage of 13.0 ± 1.0%. Four different methods—total phenolic content (TPC), ferric reducing antioxidant power (FRAP), oxygen radical absorbance capacity (ORAC), and 1,1-diphenyl-2-picryl-hydrazyl (DPPH)—were used to assess the antioxidant activity of *F. spiralis*. The bioaccessibility of the antioxidants studied, ranged between 42.7% and 59.5%, except the bioaccessibility of polyphenols in freeze-dried SW (23.0% ± 1.0%), suggesting that the freeze-drying process reduces the bioaccessibility of these compounds.

## 1. Introduction

Seaweeds (SW) have been part of the Asian diet since the prehistoric times but have been mostly overlooked in the Western world [1]. Nevertheless, France has succeeded to introduce seaweed into the European cuisine by implementing adequate legislation [1,2,3]. It is well known that marine resources are an important part of a balanced diet and contribute to a good nutritional status, mainly due to their high levels in many important nutrients that are not commonly found in other food resources. For example, SW are a good source of biological active phytochemicals, such as carotenoids, fatty acids, polysaccharides, and vitamins (A, C, D, and E) [4]. Therefore, there has been an increased interest in SW and derived products by European consumers and consequently edible seaweeds have been prepared and marketed worldwide [4,5].

Examples of secondary metabolites produced by seaweeds are fatty acids, halogenated compounds (e.g., haloforms), halogenated alkanes, alkenes, alcohols, aldehydes, hydroquinones, and ketones, carbonyls, sulfur-containing heterocyclic compounds (e.g., sulfated polysaccharides), and phenolic compounds, including quinones, flavones, flavonoids, flavonols, and phlorotannins. These secondary metabolites have been shown to have, for example, antioxidant and antimicrobial properties [6,7].

Particularly, brown SWs are especially rich in these metabolites, notably in antioxidants. These antioxidants are produced by response to the stress caused by intense sun exposure and can be effective to counterbalance the effect of reactive oxygen species (ROS), like hydrogen peroxide (H_2_O_2_), nitric oxide (NO), superoxide anion (O_2_^−^), and hydroxyl radical (OH**˙**) in the human body. ROS are produced during the cellular metabolism and are considered to be toxic, since they are highly reactive and tend to initiate chain reactions that can damage proteins, lipids, and DNA [6,8]. This negative impact can then lead to serious health problems, such as cancer, neurodegenerative diseases, and some cardiovascular diseases [9].

From these metabolites, phlorotannins are an interesting group of phenolic compounds that are normally found in high amounts in brown SW [10]. They have a wide range of molecular sizes (400–400,000 Da) and are not found in terrestrial plants. In addition to antioxidant properties, phlorotannins have a protective action against vascular diseases, anticancer activities, antihypertension activities, anti-inflammatory properties, and antimicrobial activity against food-borne bacteria, which attracted great interest from researchers and the food industry [6,11].

SW are also a source of polyunsaturated fatty acids (PUFAs) like omega-3 (for instance, eicosapentaenoic acid (EPA, 20:5ω-3), stearidonic acid (SDA, 18:4ω-3), α-linolenic acid (LNA, 18:3ω-3)), and omega-6 (arachidonic acid (ARA, 20:4ω-6)), some of which are considered essential nutrients to humans [12]. Although human beings can produce some fatty acids using the acetyl-CoA pathway, they cannot synthesize adequate amounts of omega-3 and omega-6 PUFAs, which is why it is important to introduce them in the diet. PUFAs, like EPA and docosahexaenoic acid (22:6ω-3), help control many heart related diseases. Recent studies indicate that anti-inflammatory and insulin-sensitizing effects of these fatty acids in metabolic disorders [12]. Some PUFAs were also shown to enhance the fucoxanthin anti-obese and anti-diabetic activities in mice [13].

Knowledge about benefits associated to SW consumption is of major importance to consumers and producers. However, the concentration of a compound is not enough when assessing if a nutrient is beneficial to health, since the total amount of a nutrient does not reflect its bioaccessibility. The bioaccessibility of a compound depends on the physical properties of the food matrix and special physiological conditions of the consumer, like age and health [14,15]. Only a portion of the components released during digestion (bioaccessible fraction) will be available for body physiologic functions or storage. Consequently, the real available content to the human organism can be different from the ingested. However, the bioaccessibility of bioactive compounds is still poorly understood for a realistic benefit evaluation. In fact, so far, most of the performed studies only evaluated the concentration of some of the target compounds, neglecting bioaccessibility [1,16]. Therefore, the aim of this work was to assess the bioaccessibility of polyphenols and fatty acids by simulating the human digestive process, and the antioxidant activity before and after the digestion in order to contribute for a better knowledge of the benefits associated with *Fucus spiralis* consumption. The effect of the freeze-drying (FD) process on bioaccessibility of polyphenols and antioxidant activity was also studied. Additionally, *F. spiralis,* collected from the Portuguese coast, was characterized in terms of lipid and protein content.

## 2. Materials and Methods

### 2.1. Collection and Preparation of Fucus Spiralis

*F. spiralis* was harvested in July 2015 on the north coast of Peniche, Portugal (39°37′03.53′′N 9°38′87.59W), covering the whole beach, approximately 1 km of shoreline. SW collection and identification was performed by André Horta, a marine biologist. The SW were immediately transported to the laboratory and washed with seawater to remove invertebrates and other organisms, sands and debris. In total, around of 30 kg of seaweed were collected, forming a pool sample. The samples were divided in several bags and stored at −80 °C and a portion was then freeze dried (FD) for 48 h at −60 °C (Scanvac Cool Safe, LaboGene, Lillerød, Denmark).

### 2.2. In Vitro Digestion Model

The bioaccessibility of antioxidants (both activity and phenolic content) and lipid content in fresh and FD *F. spiralis* was studied by using an in vitro method adapted from Afonso and co-workers [17]. This method includes three steps, simulating the digestive processes in the mouth, stomach, and small intestine. The composition of digestive juices (saliva, gastric, duodenal and bile) was the same described by Afonso et al. [17]. The solution attained after simulated digestion was centrifuged at about 2750× *g* for 5 min in order to separate the non-digested from the bioaccessible fraction. The antioxidant activity, total phenolic, and fatty acid contents were then analyzed in the bioaccessible fraction.

### 2.3. Calculation of Bioaccessible Lipids, Fatty Acids, Polyphenols and Antioxidant Activity

The percentage (%) of nutrients/antioxidant activity in the bioaccessible fraction was estimated as follows:(1)$$\%\;N\;\mathrm{bioaccessible}\;=\frac{N\;\mathrm{bioaccessible}}{\left[N\right]\mathrm{before}\;\mathrm{the}\;\mathrm{digestion}}\times 100$$
where [N] is the concentration of the nutrient/antioxidant activity in the fresh or FD seaweed.

### 2.4. Crude Protein

The protein level in *F. spiralis* was determined using a FP-528 DSP LECO nitrogen analyzer (LECO, St. Joseph, MI, USA), calibrated with EDTA, according to the Dumas method [18].

### 2.5. Total Lipids

Total lipids in SW samples were determined following the Folch extraction method using a mixture of chloroform and methanol (2:1, *v*/*v*) [19]. For the determination of total lipids in the bioaccessible fraction, a slight modification of the Bligh and Dyer methodology [20] was used: 4 mL of chloroform were added to the bioaccessible fraction, followed by 1 min homogenization in a vortex and then centrifugation at 2000× *g* for 5 min. The upper phase was rejected, and 4 mL of chloroform and 2 mL of water were added to the lower phase. The mixture was homogenized for 1 min in a vortex and then centrifuged at 2000× *g* for 5 min at 4 °C. The upper phase was rejected and the previous operation was repeated in the lower phase. The organic phase was then filtered through a filter containing anhydrous sodium sulphate and then evaporated in a rotary evaporator. The lipid samples were weighed, solubilized in chloroform, and stored at −20 °C until further analysis.

### 2.6. Fatty Acids

Fatty acid methyl esters (FAME’s) were prepared by acid-catalyzed transesterification using the methodology described by Bandarra et al. [21]. Samples were injected into a Varian Star 3800 CP gas chromatograph (Walnut Creek, CA, USA and equipped with an auto sampler with a flame ionization detector at 250 °C. FAME’s were identified by comparing their retention times with those of Sigma–Aldrich standards (PUFA-3, Menhaden oil, and PUFA-1, Marine source from Supelco Analytical).

### 2.7. Total Phenolic Content and Antioxidant Capacity

#### 2.7.1. Preparation of Seaweed Extract and Bioaccessible Factions

Seaweed extracts were prepared according to the method adapted from Pinteus et al. [10]. SW samples (4 g of wet SW or 1 g of FD SW) were mixed with methanol (6 mL) and stirred for 30 min. After centrifugation at 3214× *g* for 10 min (Eppendorf, centrifuge 5810 R, Hamburg, Germany), the supernatant was collected and filtered through a Büchner funnel. The process was repeated 3 times. The solvents were evaporated, and the extracts were preserved at −20 °C in dimethyl sulfoxide (DMSO).

In order to be able to analyze the antioxidants in the bioaccessible fraction, 10 g of each bioaccessible fraction were freeze-dried (during 48 h at −60 °C) and the resulting mass was dissolved in DMSO and stored at −20 °C.

#### 2.7.2. Analysis of Total Phenolic Content (TPC)

The TPC of *F. spiralis* extract was determined using the Folin–Ciocalteu method adapted to microscale with minor modifications as described by Pinteus et al. [10]. The TPC is expressed as mmol equivalents of gallic acid per gram of SW.

#### 2.7.3. DPPH (1,1-Diphenyl-2-Picryl-Hydrazyl) Radical Scavenging Activity

DPPH radical scavenging activity was performed as described by Pinteus et al. [10]. Results are expressed as mean values ±SD (standard deviation). EC_50_ values (μg/mL) were also determined for the extracts which scavenged DPPH radical over 50 %.

#### 2.7.4. Oxygen Radical Absorbance Capacity (ORAC)

Oxygen Radical Absorbance Capacity (ORAC-fluorescein) assay was performed as described by Pinteus et al. [10]. The results were expressed in mmol equivalents of Trolox/g of SW.

#### 2.7.5. Ferric Reducing Antioxidant Power (FRAP) Assay

The FRAP assay was performed according to the method of Benzie and Strain with slight modifications [22]. The FRAP assay measures the ability of the antioxidants to reduce ferric-tripyridyl-triazine (Fe^3+^-TPTZ) complex to the blue colored ferrous form (Fe^2+^) which absorbs light at 593 nm. Briefly, standard or sample extract (10 µL) were mixed with Fe^3+^-TPTZ reagent (300 µL acetate buffer 300 mM, pH 3.6; 10 mM TPTZ in 40 mM HCl; 20 mM FeCl_3_·6H_2_O; 10:1:1 (*v*/*v*/*v*)) and poured into microplate wells. The plate was incubated at 37 °C for the duration of the reaction. The absorbance readings were taken at 593 nm after 30 min reaction using the microplate reader (Synergy H1 Multi-Mode Microplate Reader, BioTek^®^ Instruments, Winooski, VT, USA). Ascorbic acid was used as the control (20–1000 µM). The results are expressed in mmol equivalents of ascorbic acid/g of SW.

### 2.8. Statistical Analysis

All measurements were performed in triplicate. All data were checked for normality (Kolmogorov-Smirnov’s test) and homoscedasticity (Levene’s F-test). For antioxidant content, the initial (fresh and FD SW) and respective bioaccessible fractions were compared using the parametric Student’s *t*-test for independent samples [23]. In addition, differences in bioaccessibility percentages between the antioxidant activity (FRAP and ORAC) and TPC on fresh and FD samples were evaluated by a two-way analysis of variance (ANOVA) [23]. Whenever applicable, Bonferroni’s multiple comparison tests were performed to determine differences in bioaccessibility among antioxidant methods and/or fresh and FD samples [23]. Results were expressed as mean ±standard deviation (SD). For all statistical tests, the significance level (α) was set at *p*-value ≤ 0.05. Calculations were performed using IBM SPSS Statistics 22 (Armonk, NY, USA). The half maximal inhibitory concentration (IC50) was calculated from nonlinear regression analysis using the software GraphPad InStat v5.1 (San Diego, CA, USA) with the equation [24]:(2)$$Y=100/\left(1+10^{\left[X-\mathrm{Log}\left({\mathrm{IC}}_{50}\right)\right]}\right)$$

## 3. Results and Discussion

### 3.1. Protein Content

The *F. spiralis* protein content of 12.73 ± 1.38% DW is within the ranges described for brown algae (3%–15% of DW) [25,26]. However, Paiva and co-workers reported lower values for *F. spiralis* collected in Azores islands (Portugal) (4.14% and 8.25%) for samples collected in the same time of the year, which might explain this variation [27]. Nevertheless, protein content varies with seawater temperature, salinity, nutrients, and geographical location [25,27]. Moreover, this protein content is also comparable to that of some grains like oats (13.4%), wheat (13.8%) and is higher than that of corn (9.4%) and rice (7.1%) [28]. Nonetheless, SW consumption is typically very low, making this a minor contribution to the overall protein content of a healthy diet. The remaining chemical composition both ash and moisture were already characterized on a previous work [29].

### 3.2. Lipid and Fatty Acids Content

SW have significantly lower lipid content than marine fish. For instance, oily fish have been reported to contain between 20%–50% DW while SW contain 1%–5% total lipids DW [12]. Nevertheless, brown SW lipids comprise many types of bioactive compounds, such as omega-3 PUFAs, omega-6 arachidonic acid, fucoxanthin, fucosterol, and some polyphenols that, when introduced into a healthy diet, can become an added value for the consumer [12].

In this work, the total lipid content obtained was 3.49% ± 0.30% DW for freeze-dried SW and 0.82% ± 0.07% wet weigh for the fresh SW, which is in accordance with the values reported for SW in general [12]. Paiva et al. attained a different result with *F. spiralis* (from Azores), obtaining a higher lipid content of 5.23% ± 0.03% DW [26]. However, they collected the SW in January (average temperature of 16 °C) while the SW studied in the present work were harvested in July (average temperature of 23.18 °C). This lipid accumulation can be explained with the cessation of growth caused by the cold stress, which leads to the incorporation of these compounds into the cell membranes to retain the membrane fluidity [30,31].

Furthermore, SW lipid content also depends on the species and especially on the geographical location, which will influence temperature, salinity levels and different light intensity to which SW are exposed [12,32]. In addition, it has been reported that some tropical species showed significantly lower lipid content than cold-water species [33,34]. It is also known that brown SW that grow in low temperatures, normally, accumulate more omega-3 and omega-6 PUFAs. The major omega-3 PUFAs found in brown SW are eicosapentaenoic acid, stearidonic acid, and α-linolenic acid while arachidonic acid is the major omega-6 PUFA [12,35,36,37,38,39]. In this work, for the fatty acid group, PUFAs represented 37.87%, monounsaturated fatty acids (MUFAs) 29.18%, and saturated fatty acids (SFA) 32.87% (Table 1). These results agree with what has been described for this seaweed species. For instance, Paiva et al. reported 38.97% of PUFAs, 37.10% of MUFAs, and 33.59% of SFA [26]. Eicosapentaenoic acid was the predominant omega-3 with 7.45% ± 0.05% and arachidonic acid was the major omega-6 with 16.38% ± 0.04%. In contrast, although Paiva et al. reported lower contents of arachidonic acid (0.43%) and eicosapentaenoic acid (1.05%) in SW collected in January, they described similar contents of palmitic acid and oleic acid (18.77% and 21.04%, respectively). On the other hand, Campos et al. [40] noted that the previous study might wrongly idenitfy eicosapentaenoic acid, its true value being from 20:3 ω3 with 16%, which is in agreement with our obtained results. However, in the Paiva et al. study, SWs were dried in an oven (65 °C); in our study, SWs were freeze-dried. This difference caused different degrees of oxidation between the two samples, explaining the different results [26]. Similarly, they concluded that arachidonic acid content was higher in samples collected in July (16.4%) than in those collected in January (12.2%), while the eicosapentaenoic acid content was higher in January (13.7%) than in July (9.2%). Kayama et al. reported similar results for *Porphyra yezoensis*, with seaweeds grown in colder waters (10 °C) presenting higher levels of eicosapentaenoic acid than those grown at higher temperatures (20 °C) [34]. These studies reinforce the seaweed adaptive mechanism when exposed to different conditions. Further studies on seasonal variation of lipids and fatty acids of the *F. spiralis*, from Peniche coast, are therefore needed to better understand the results.

### 3.3. Antioxidant Activity

Due to the presence of different bioactive compounds with anti-oxidative potential in the crude extracts of samples, many different methods have been used to investigate antioxidant activity in recent years. In this study, the antioxidant activity and polyphenol content were assessed by FRAP, ORAC, and TPC, respectively, and the results obtained before in vitro digestion are presented in Table 2.

Regarding the DPPH assay, FD seaweed (initial extract) presented an IC_50_ of 0.917 ± 0.122 mg SW/mL.

As previously mentioned, *F. spiralis* was collected in the intertidal region, which is exposed to the air and sun during low tide and submerged by the rising tide. Because of the submersion alternation and air exposure, marine organisms are forced to endure extreme conditions, such as salinity, dryness, ultraviolet (UV) radiation, and hydrodynamics among others. As a stress response mechanism to these extreme conditions, namely UV radiation, SW produced a high range of antioxidant compounds, including polyphenols. Several studies have shown that all classes of SW produce compounds with antioxidant activity [41]. The algae of the genus *Fucus* are known to be constituted mainly by non-digestible polysaccharides and rich in polyphenols [10]. These polyphenols are highly complex and their antioxidant power can be up to 100 times stronger than the biophenols produced from the terrestrial plants [7]. Thus, SW are a promising natural alternative to substitute some synthetic antioxidants like butylated hydroxytoluene (BHT) that raise some health concerns [42,43].

In a previous work, Pinteus et al. [10] evaluated the TPC and the antioxidant activity of 10 seaweeds from the Peniche coast and concluded that *F. spiralis* and *Saccorhiza polyschides* had the highest antioxidant activity and total phenolic content.

Following up with this work, *F. spiralis* presented a high total phenolic content (0.049 ± 0.005 mmol EGA/g DW) when compared with *F. spiralis* collected in Denmark (0.044 ± 0.001 mmol EGA/g DW), and much more than in Scotland (0.014 ± 0.000 mmol EGA/g DW) [44,45]. The results obtained with the FRAP assay (31.47 ± 0.52 mmol EAA/g DW) agree with the results obtained through the other methods, giving its high total phenolic content, and also higher than that found for Scotland SW (18.8 ± 0.7 mmol E of Trolox/g DW) [45]. These differences might be due to the geographic differences and consequently climatological variances. Portugal has in general higher temperatures and sun exposure than Scotland and Denmark, which will trigger the seaweeds to produce more antioxidants compounds for their protection [46,47].

DPPH radical scavenging method is commonly used to evaluate the antioxidant capacity of the seaweed extracts, because of its reliability. A previous study of Andrade and co-workers with *F. spiralis*, collected from Peniche coast, reported an IC_50_ of 2.49 mg DW/mL for the DPPH radical scavenging activity assay, which is higher than the value obtained in this work (0.917 mg DW/mL) [48]. This difference may be explained by the different harvesting season. The SW in Andrade’s study were collected in September while in this study they were collected in July, having the latter suffered higher sun exposure and consequently higher oxidative stress.

Overall, the high antioxidant activity observed can be explained by the presence of mannitol, fucosterol, and phlorotannin reported for *F. spiralis* from Peniche coast in the most recent studies [10,48]. Mannitol is a known free radical scavenger, and it is used as reference in the evaluation of the activity against hydroxyl radical in different assays, which helps explaining the high ORAC and DPPH values [4]. Fucosterol has also been reported to have antioxidant activity with special interaction with the TPC method [6,49]. Phlorotannins are known to give strong responses with the Folin-Ciocalteu reagent, therefore explaining the high TPC value obtained [10]. Besides the ones described, other compounds present in the seaweed may be responsible for the activities observed [50].

### 3.4. Bioaccessibility

As mentioned earlier, *F. spiralis* is a possible source of nutrients such as polyphenols and other minor phytochemicals that are considered bioactive compounds, but the concentration of a compound in food is not enough for assessing its beneficial or hazardous influence on health since the total amount may significantly differ from the amount that is bioaccessible. The bioaccessibility is the fraction of a specific compound that is released from the food matrix during digestion, which is then available for intestinal absorption [1,16]. Bioaccessibility is a recent notion which appeared in the 1990s as scientists realized that the food matrix influences how the compounds are absorbed in the human body. Thus, the information related to bioaccessibility of nutrients and antioxidant compounds from seaweeds is still scarce.

Regarding the results obtained for the total lipid, the bioaccessibility was 12.1% ± 0.1% FD SW. The lipid bioaccessibility for fresh SW wasn’t evaluated due to the low lipid content. Regarding bioaccessibility of the major fatty acids, although EPA presented the highest bioaccessibility percentage, it was a very low value of 13.0% ± 1.0% (Table 1).

The poor lipid bioaccessibility can be related to the fact that most of the lipid content in SW are in form of phospholipids and glycolipids, both molecules associated with cell membranes, which, in turn, are tightly protected by the cell walls. In fact, the SW did not suffer substantial physical disintegration during the digestion, possibly as a result of the high fiber content in their cell wall. In addition, the human gastrointestinal tract does not produce the required degradation enzymes to metabolize these polysaccharides resulting in a poor lipid release [12,51]. For these reasons, the seaweed *F. spiralis,* in the form it was analyzed, might not be considered a good source of lipids and consequently fatty acids. Nevertheless, these results add some information to this still poorly studied nutritional field [32].

Concerning the antioxidants (both activity and total polyphenols) bioaccessibility, the results obtained before and after in vitro digestion are presented in Table 2. It can be verified that there is a significant reduction of the antioxidant activity after in vitro digestion. Regarding the DPPH assay, the bioaccessible fraction of the seaweed product had a very low activity, making it impossible to determine IC_50_. In terms of the bioaccessibility percentage (Table 2), the TPC bioaccessibility for FD SW was the lowest value observed, with 22.4 ±1.0% (2 way ANOVA, Bonferroni test; *p* < 0.05), when compared with the bioaccessibility for TPC of fresh SW and with antioxidant capacity (assessed by ORAC and FRAP) for the same sample. The remaining bioaccessibility percentages oscillated between 42.7% and 59.5%.

Overall, there is very scarce information regarding SW total polyphenols bioaccessibility and antioxidant capacity obtained after in vitro digestion and the research that exists is either very compound specific or about the use of SW as raw-material in other products [52]. Although studies that evaluate bioaccessibility of antioxidants from SW are few, there are already a couple of studies with vegetables and fruits. Kamiloglu et al. studied the antioxidant activity of black carrots at the bioaccessible fraction. The values reported for TPC ranged from 30% to 40% [53], which is in accordance with the results obtained. Tagliazucchi et al. also reported similar results for grapes, with 49.7% TPC bioaccessibility and 55.5% of antioxidants activity maintenance in the bioaccessible fraction for the FRAP assay [54]. On the other hand, in a study with different types of apples [55], the TPC bioaccessibility values were higher (75%). This difference may be explained with the higher fiber content present in *F. spiralis* (around 60%) that may difficult the release of the antioxidants during digestion [52]. In addition, the fact that SW produce other types of antioxidant compounds, like phlorotannins, non-existent in terrestrial plants, also contributes to these differences. Moreover, different interactions with the digestive system and the antioxidants methods are to be expected.

The significant reduction of the polyphenols bioaccessibility attained for the FD SW may be the result of several factors. The amino acids and purines released from the actions of proteases and ribonucleases used in the in vitro digestion procedure may interfere with the Folin–Ciocalteu method by reacting with the Folin–Ciocalteu reagent, being more problematic on FD samples due to their concentrated state. The fact that the antioxidant activity of polyphenols is strongly pH-dependent may also have a great impact in this sample, due to the high ion concentration resulting from the freeze-drying process. Moreover, phenolic compounds can also interact with dietary compounds released during digestion, such as dietary fiber, minerals, or proteins, affecting solubility and bioaccessibility [53,54,56,57].

Nevertheless, the conjunction of the high initial antioxidants and the bioaccessible values reinforces the nutraceutical potential of *F. spiralis*. Even though this compounds’ bioavailability is presumably low, they still take a protective effect on the intestine [58]. This is of special importance to a country where intestinal related cancers are the third most prevalent type of cancer [59] and thus the introduction of *F. spiralis* in the diet may help reduce the disease’s impact.

## 4. Conclusions

Although the lipid content in *F. spiralis* is low, compared to other marine sources, up to 40% are polyunsaturated fatty acids. On the other hand, the lipids bioaccessibility was very low for the conditions accessed, especially when comparing with the antioxidant bioaccessibility.

The antioxidant content/activity of *F. spiralis*, especially in the freeze-dried form, may be valuable to the pharmaceutical and food industries. Indeed, it appears to be indifferent, regarding the antioxidant bioaccessibility, to consume fresh SW or freeze-dried SW, except for the lower polyphenol content in the FD SW. However, further research is required for a better understanding on how digestion affects antioxidants in their structure, function, and role, as well as how they interact with the intestine to be absorbed by the human organism.

## Figures and Tables

**Table 1 foods-09-00440-t001:** Major fatty acids (FA) detected in freeze-dried (FD) *F. spiralis* (% of total fatty acids) and major fatty acids bioaccessibility (Bioac) percentages from freeze-dried *F. spiralis*. Values are mean ± SD.

Fatty Acids	*(% of total FA)*		*Bioac (%)*
Myristic acid (C14:0)	10.92	±0.30	
Palmitic acid (C16:0)	16.25	±0.10	
Oleic acid (C18:1ω9)	26.39	±0.27	
Linoleic acid (C18:2ω6)	5.51	±0.05	
α-linolenic acid (C18:3ω3)	4.56	±0.03	10.9 ± 1.6
Stearidonic acid (C18:4ω3)	2.52	±0.02	8.1 ± 0.9
Arachidonic acid (C20:4ω6)	16.38	±0.04	6.1 ± 0.8
Eicosapentaenoic acid (C20:5ω3)	7.45	±0.05	13.0 ± 1.0

Σ Saturated	32.37	±0.21	
Σ Monounsaturated	29.18	±0.24	
Σ Polyunsaturated	37.37	±0.03	
Σ ω3	15.07	±0.01	
Σ ω6	22.46	±0.02	
*ω3/ω6*	0.67	±0.00	
Unidentified	1.2	±0.23	

**Table 2 foods-09-00440-t002:** Total phenolic content (TPC) and antioxidant activity of fresh (F) and freeze-dried (FD) seaweed: ferric reducing antioxidant power (FRAP) and oxygen radical absorbance capacity (ORAC) before (initial) and after in vitro digestion (bio), and their bioaccessibility percentages (bioac). Values are presented as average ± standard deviation; fresh seaweed (F) results are presented in wet-weight and FD are in dry-weight. * Statistic differences comparing the initial values and the in vitro digestion values (*p* < 0.05). ^#^ Represents bioaccessibility value statistically different between seaweed condition (fresh and freeze-dry) and between antioxidant method within each condition (*p* < 0.05).

	TPC		
	Initial	Bio	Bioac
	(mmol EGA/g)	(mmol EGA/g)	(%)
**F**	0.016 ± 0.002 *	0.007 ± 0.001 *	43.7 ± 6.2
**FD**	0.049 ± 0.005 *	0.011 ± 0.001 *	22.4 ± 1.0 ^#^
	**FRAP**		
	initial	Bio	Bioac
	(mmol EAA/g)	(mmol EAA/g)	(%)
**F**	17.65 ± 0.51 *	9.05 ± 1.20 *	51.3 ± 6.8
**FD**	31.47 ± 0.52 *	18.73 ± 0.63 *	59.5 ± 2.0
	**ORAC**		
	initial	Bio	Bioac
	(mmol E Trolox/g)	(mmol E Trolox/g)	(%)
**F**	104.95 ± 0.80 *	44.82 ± 5.71 *	42.7 ± 5.4
**FD**	304.09 ± 17.81 *	138.69 ± 40.44 *	45.6 ± 13.3

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
