# Peer review of "Bioaccessibility of Antioxidants and Fatty Acids from Fucus Spiralis"

_foods, 2020, doi:10.3390/foods9040440_

Round 1

Reviewer 1 Report

The quality of the manuscript “Bioaccessibility of antioxidants and fatty acids from Fucus spiralis” submitted for publication in Foods is quite good. The research topic is interesting and the results could have interest for the food industry. However, only the antioxidant capacity of the different extracts of SW, studied by different methods, is given whereas the chemical composition and description is completely missing. Given the potential benefice of this type of food for human diet and health, the exact polyphenols profile should be analyzed by LC/MS and lipids profile by GC/MS. This supplemental work could improve the quality of the submitted article.

Author Response

Point 1: The quality of the manuscript “Bioaccessibility of antioxidants and fatty acids from Fucus spiralis” submitted for publication in Foods is quite good. The research topic is interesting and the results could have interest for the food industry. However, only the antioxidant capacity of the different extracts of SW, studied by different methods, is given whereas the chemical composition and description is completely missing.

Response 1:  Thank you for your general comments. The chemical composition that is not presented and discussed in this article (moisture and ask content) has been previously characterized/discussed in a previous work as stated on section 3.1:

Francisco, J.; Cardoso, C.; Bandarra, N.; Brito, P.; Horta, A.; Pedrosa, R.; Gil, M.M.; Delgado, I.M.; Castanheira, I.; Afonso, C. Bioaccessibility of target essential elements and contaminants from Fucus spiralis. J. Food Compos. Anal. 2018, 74, 10–17.

However, the objective of this work was to study the effect of freeze-dry and digestive process on bioaccessibility of target compounds and bioactivities, i.e., antioxidant activity, total polyphenols, lipid, and fatty acids content because of their potential use to the food industry, and not assess the chemical composition.

Point 2: Given the potential benefice of this type of food for human diet and health, the exact polyphenols profile should be analyzed by LC/MS and lipids profile by GC/MS.

Response 2: Thank you for your comments regarding the LC/MS. The authors agree it could be of interesting note to evaluate these seaweed (SW) extracts especially the molecules that end up being bioaccessible. Nevertheless, the aim of the work was to have an overall understanding regarding the suitability of this SW as promising nutraceutical and set as groundwork for future studies. As referred, scarce information is available on how the SW behaves during the digestive process and how the freeze-drying process might affect the bioaccessibility of the different compounds. Nevertheless, the authors are considering doing it for our next works that will also go in more depth on how we can process this SW or integrate it in other food matrix as a complement. Regarding the utilization of a GC/MS the authors believe that it wouldn’t be necessary due to the low % of unidentified molecules (1.2%).

Reviewer 2 Report

In my opinion the work have been performed carefully.

Congratulations for the research paper.

Author Response

Point 1: In my opinion the work have been performed carefully. Congratulations for the research paper.

Response 2: Thank you very much for your kind words.

Reviewer 3 Report

Interesting paper because it offers detailed data on the chemical composition of a population of Fucus cf. spiralis and treats the aspect of the bioaccesibility of its active compounds, issue many times not taken into account. However, this work has many basic problems. In the first place it is necessary to indicate that Fucus species due to their texture and organoleptic characteristics are not consumed directly as food, although locally part of their thalli (receptacles) may now be by fits and starts consumed. Traditionally Fucus species only have medicinal uses (infusions, powder in capsules, etc.). Moreover, brown algae of the order Fucales and Laminariales have high levels of iodine, so their consumption can also be a problem for some people. On the other hand, Fucus species are of a very complex taxonomy, it is very difficult to make comparisons of the results obtained in this work with the data of others "F. spiralis" from differents regions of the North Atlantic, very probably not coespecific. In fact, it is very likely that what is identified in this paper as Fucus spiralis was actually F. guiryi. In other words, this paper does not prove reliably with which Fucus species has been worked on.

On the other hand nothing is said about the phenological state of the population of Fucus studied. The richness in fiber, fatty acids, carotenoids and other substances in the same population of Fucus depends greatly on their phenological state since these compounds are especially abundant in mature receptacles. It would have been interesting to do this study in the same population at different times of the year to see its real variability, since comparisons of the data obtained punctually in this work with those of other authors in other regions, times of the year and, perhaps, over other species does not make much sense.

It is true that seaweeds have polyunsaturated fatty acids (PUFAs), but in a very small amount (the same authors recognize it in line 183, where they indicate that seaweeds have between 1-5% total lipids, of which PUFAs are only 37 % and its bioaccessibility is only 12% ...), so the contribution of PUFAs by Fucus consumption, which is never going to be high, is practically negligible.The same authors end up in lines 298-299 that “F. spiralis might not be considered a good source of lipids and consequently fatty acids.

On the other hand, it is necessary to indicate that fucaceae are algae that, when dried in the air, suffer significant alterations in their composition, some of their compounds oxidize and remain black, a process that does not occur in freeze dried seaweeds. Perhaps this can explain many of the analytical differences obtained between dry biomass in an oven or by freeze dried.

With respect to the calculation of the percentage of proteins, estimates based on the richness in N are not very reliable and the Dumas method provided results that were relatively higher than the Kjeldahl method, but the difference between the methods depended on the type of foodstuff. The authors say that the protein content of Fucus cf. spiralis is comparable to that of soybeans, for which they give a percentage of 13.0%, when the bibliography usually indicates values of the order of 35% ... Fucus consumption will always be in very small quantities, so its contribution in proteins it will also be practically negligible.

Other comments:

Line 37. It is very debatable that seaweeds have vit. B12. Seaweeds seem to have structural analogues to vit. B12, but they do not have their catalytic function.

Line 210. Replace: “M.K. Kim, Dubacq, Thomas, & Giraud” by “Kim and co-workers”

Line 214. “thalli” is not a specific epithet, is the plural of thallus!

Author Response

Point 1: In the first place it is necessary to indicate that Fucus species due to their texture and organoleptic characteristics are not consumed directly as food, although locally part of their thalli (receptacles) may now be by fits and starts consumed.

Response 1: It is, indeed, a valid point. Nonetheless the aim of this work was to have results on the effect of freeze-dry and digestive process on bioaccessibility of target compounds and bioactivities, such as, antioxidant activity, total polyphenols, lipid, and fatty acids content because of their potential as nutraceutical and ingredient in dried food formulations and if it is worth, or not, the bioprospecting of compounds for future studies and application, as for nutraceutical. As future work, the authors aim to see how SW can be integrated in the diet or food product to enhance some of its proprieties. The authors believe that this study is of major importance for further development of functional foods based on marine-derived ingredients

Point 2: Moreover, brown algae of the order Fucales and Laminariales have high levels of iodine, so their consumption can also be a problem for some people.

Response 2: This is a relevant and interesting subject. In fact, the team has already dedicated to this issue and published an article where it explores the results obtained, also introducing the bioaccessibility approach (Francisco, J.; Cardoso, C.; Bandarra, N.; Brito, P.; Horta, A.; Pedrosa, R.; Gil, M.M.; Delgado, I.M.; Castanheira, I.; Afonso, C. Bioaccessibility of target essential elements and contaminants from Fucus spiralis. J. Food Compos. Anal. 2018, 74, 10–17). Furthermore, taking into account what is known from countries with relatively high consumption levels of seaweed, such as Japan, and the uncertain iodine bioaccessibility reported for other groups of seaweeds (Afonso, C., Cardoso, C., Ripol, A., Varela, J., Quental-Ferreira, H., Pousão-Ferreira, P., Ventura, M.S., Delgado, I.M., Coelho, I., Castanheira, I. and Bandarra, N.M. 2018. Composition and bioaccessibility of elements in green seaweeds from fish pond aquaculture. Food Research International, 105: 271-277.), the problem of excessive dietary iodine is conjectural, thus warranting further research.

Point 3: On the other hand, Fucus species are of a very complex taxonomy, it is very difficult to make comparisons of the results obtained in this work with the data of others "F. spiralis" from differents regions of the North Atlantic, very probably not coespecific. In fact, it is very likely that what is identified in this paper as Fucus spiralis was F. guiryi. In other words, this paper does not prove reliably with which Fucus species has been worked on.

Response 3: F. spiralis was harvested in July 2015 in the north coast of Peniche, Portugal (39°37'03.53''N 9°38'87.59W) where Fucus guiryi has never been identified.

Besides this, SW were identified by André Horta (marine biologist) and Rui Pedrosa (biochemical), experts in SW taxonomy with several publications in this field:

  1. Francisco, J., Cardoso, C., Bandarra, N., Brito, P., Horta, A., Pedrosa, R., Gil, M.M., Delgado, I.M., Castanheira, I., Afonso, C. (2018). Bioaccessibility of target essential elements and contaminants from Fucus spiralis. Journal of Food Composition and Analysis 74: 10-17. doi: 10.1016/j.jfca.2018.08.003
  2. Pinteus, S.; Silva, J.; Alves, C.; Horta, A.; Thomas, O. P.; Pedrosa, R. Antioxidant and cytoprotective activities of fucus spiralis seaweed on a human cell in vitro model. Int. J. Mol. Sci. 2017, 18.
  3. Pinteus, S.; Silva, J.; Alves, C.; Horta, A.; Fino, N.; Rodrigues, A. I.; Mendes, S.; Pedrosa, R. Cytoprotective effect of seaweeds with high antioxidant activity from the Peniche coast (Portugal). Food Chem. 2017, 218.
  4. Alves, C.; Pinteus, S.; Horta, A.; Pedrosa, R. High cytotoxicity and anti-proliferative activity of algae extracts on an in vitro model of human hepatocellular carcinoma. Springerplus 2016, 5.
  5. Alves, C.; Pinteus, S.; Simões, T.; Horta, A.; Silva, J.; Tecelão, C.; Pedrosa, R. Bifurcaria bifurcata: a key macro-alga as a source of bioactive compounds and functional ingredients. J. Food Sci. Technol. 2016, 51, 1638–1646.
  6. Rodrigues, D.; Alves, C.; Horta, A.; Pinteus, S.; Silva, J.; Culioli, G.; Thomas, O. P.; Pedrosa, R. Antitumor and antimicrobial potential of bromoditerpenes isolated from the Red Alga, Sphaerococcus coronopifolius. Drugs 2015, 13.
  7. Pinteus, S.; Alves, C.; Monteiro, H.; Araújo, E.; Horta, A.; Pedrosa, R. Asparagopsis armata and Sphaerococcus coronopifolius as a natural source of antimicrobial compounds. World J. Microbiol. Biotechnol. 2015.
  8. Horta, A.; Pinteus, S.; Alves, C.; Fino, N.; Silva, J.; Fernandez, S.; Rodrigues, A.; Pedrosa, R. Antioxidant and antimicrobial potential of the Bifurcaria bifurcata epiphytic bacteria. Drugs 2014, 12, 1676–89.

Besides this, aiming at clarifying this topic, the following sentence, “F. spiralis was harvested in July 2015 in the north coast of Peniche, Portugal (39°37'03.53''N 9°38'87.59W), covering the whole beach, approximately 1 km of shoreline.” was re-written:

 “F. spiralis was harvested in July 2015 in the north coast of Peniche, Portugal (39°37'03.53''N 9°38'87.59W), covering the whole beach, approximately 1 km of shoreline by André Horta, a marine biologist.”

Point 4: On the other hand nothing is said about the phenological state of the population of Fucus studied. The richness in fiber, fatty acids, carotenoids and other substances in the same population of Fucus depends greatly on their phenological state since these compounds are especially abundant in mature receptacles. It would have been interesting to do this study in the same population at different times of the year to see its real variability, since comparisons of the data obtained punctually in this work with those of other authors in other regions, times of the year and, perhaps, over other species does not make much sense.

Response 4: Authors are aware of this reality, in fact we gave it as an explanation for the differences attained (for example, lines 195 and 258). Nonetheless authors see these comparisons of results with other works as important in order to provide some framework to the attained results. Regarding the possibility of comparing SW harvested in different seasons, it was just not the aim for this particular study. Authors intended to answer 2 simple questions, to study, overall, the antioxidants compounds/fatty acids bioaccessibility of this SW and to understand how the freeze-drying process affects the bioaccessibility of these compounds. Of course, given the promising aspects of this SW and the purpose of developing future functional foods incorporating it, authors foresee future studies tackling the particular issue of the reflection of seasonality on nutritional and bioactive profile of the SW.

Point 5: It is true that seaweeds have polyunsaturated fatty acids (PUFAs), but in a very small amount (the same authors recognize it in line 183, where they indicate that seaweeds have between 1-5% total lipids, of which PUFAs are only 37 % and its bioaccessibility is only 12% ...), so the contribution of PUFAs by Fucus consumption, which is never going to be high, is practically negligible. The same authors end up in lines 298-299 that “F. spiralis might not be considered a good source of lipids and consequently fatty acids.

Response 5: The authors agree that seaweeds have polyunsaturated fatty acids (PUFAs), but in a very small amount, however the authors believe that the understanding of its bioaccessibility is important. The work presented in the manuscript intends to be a guided approach and methodology on the bioaccessibility of different SW compounds independently of the amount.

Point 6: On the other hand, it is necessary to indicate that fucaceae are algae that, when dried in the air, suffer significant alterations in their composition, some of their compounds oxidize and remain black, a process that does not occur in freeze dried seaweeds. Perhaps this can explain many of the analytical differences obtained between dry biomass in an oven or by freeze dried.

Response 6: Thank you for the comment. The authors agree and aiming at clarifying this topic, the following sentence was added to the manuscript:

“However, in Paiva work their SW where dried in an oven (65°C) in opposite in this work that where freeze-dried, which lead to different degrees of oxidation between the two samples, explain the different results.”

Point 7: With respect to the calculation of the percentage of proteins, estimates based on the richness in N are not very reliable and the Dumas method provided results that were relatively higher than the Kjeldahl method, but the difference between the methods depended on the type of foodstuff. The authors say that the protein content of Fucus cf. spiralis is comparable to that of soybeans, for which they give a percentage of 13.0%, when the bibliography usually indicates values of the order of 35%... Fucus consumption will always be in very small quantities, so its contribution in proteins it will also be practically negligible.

Response 7: There is a never-ending discussion about best method. Kjeldahl and Dumas methods are similar in their general approach, since they estimate the nitrogen content, being the protein content calculated on the basis of an adequate coefficient that is dependent on the type of foodstuff. On the other hand, the Dumas method is much more friendly to the environment and considered by many to be more accurate than the Kjeldahl method, since it is based on the measurement of thermal conductivity instead of a chain of titrations. The overall criticism of the unreliability of N-estimation methods is valid, but the reviewer must be aware that these methods have been frequently applied to seaweeds and even more frequently to other foodstuffs. There are plenty of scientific papers in indexed journals that prove this.

On the other hand, the authors totally agree with the reviewer regarding comparison to soybeans, since authors were misled by the cited reference. Hence, authors have removed the reference to the soybeans from the text. Moreover, though seaweed consumption is typically very low, it is worth assessing its main nutrients, including protein, and offer a comparative perspective by drawing on data from relevant foodstuffs. In any case, the authors agree with the reviewer and the following sentence was added to the manuscript:

“Nonetheless, SW consumption is typically very low, making this a minor contribution to the overall protein content of a healthy diet. The remaining chemical composition both ash and moisture were already characterized on a previous work.”

Point 8: Line 37. It is very debatable that seaweeds have vit. B12. Seaweeds seem to have structural analogues to vit. B12, but they do not have their catalytic function.

Response 8: Removed as suggested.

Point 9: Line 210. Replace: “M.K. Kim, Dubacq, Thomas, & Giraud” by “Kim and co-workers”

Response 9: The reference was re-written.

Pointe 10: Line 214. “thalli” is not a specific epithet, is the plural of thallus!

Response 10: The authors agree with the reviewer and the word “thalli” was removed.